# Recurrence Pattern of Left Upper Lobectomies and Trisegmentectomies: Systematic Review and Meta-Analysis

**DOI:** 10.3390/jcm14124385

**Published:** 2025-06-19

**Authors:** Borja Aguinagalde, Juan A. Ferrer-Bonsoms, Iker López, Jon Ander Lizarbe, Arantza Fernandez-Monge, Maria Mainer, Raul Embun, Jon Zabaleta

**Affiliations:** 1Osakidetza Basque Health Service, Department of Thoracic Surgery, Donostia University Hospital, 20014 Donostia, Spain; borja.aguinagaldevaliente@osakidetza.eus (B.A.); jonander.lizarbebon@osakidetza.eus (J.A.L.);; 2Department of Surgery, UPV/EHU, 48940 Leioa, Spain; 3Biogipuzkoa, Lung and Pleural Diseases Research Group, 20014 San Sebastian, Spain; 4Biomedical Engineering and Science Department, TECNUN, Universidad de Navarra, 20018 San Sebastian, Spain; 5Servicio de Cirugía Torácica, Hospital Universitario Miguel Servet, IIS Aragón, Universidad Zaragoza, 50009 Zaragoza, Spain; raulembun@gmail.com

**Keywords:** lung cancer, lobectomy, trisegmentectomy, sublobar resection, systematic review

## Abstract

**Background:** Surgical resection remains the standard treatment for early-stage non-small-cell lung cancer (NSCLC). Traditionally, lobectomy has been considered the procedure of choice; however, emerging evidence suggests that trisegmentectomy may offer comparable outcomes. This meta-analysis evaluates whether left upper lobe trisegmentectomy provides non-inferior or superior oncologic outcomes compared to left upper lobectomy, with particular attention to recurrence patterns. **Methods:** Following PRISMA guidelines, we included comparative studies evaluating left upper lobectomy versus trisegmentectomy. Outcomes assessed included recurrence (locoregional and distant), morbidity, and the length of hospital stay. A meta-analysis was conducted using the metabin function from the R meta package. **Results:** Of 14 identified articles, 9 met the inclusion criteria. No significant differences were observed in locoregional recurrence. However, distant recurrence was significantly lower in the trisegmentectomy group (OR 0.58; 95% CI 0.41–0.82). While overall morbidity showed no significant difference (OR 0.95), analysis of matched studies favored trisegmentectomy (OR 0.73; 95% CI 0.56–0.96). Hospital stay was significantly shorter in the trisegmentectomy group (OR –0.94; 95% CI –1.26 to –0.63). **Conclusions:** Trisegmentectomy and lobectomy exhibit distinct recurrence patterns, with lobectomy associated with a higher rate of distant recurrence. Trisegmentectomy may provide oncologic and perioperative advantages in appropriately selected patients. The systematic review and meta-analysis are registered in PROSPERO (registration number: CRD420251066445).

## 1. Introduction

Surgical resection remains the standard treatment for patients with early-stage non-small-cell lung cancer (NSCLC). Traditionally, lobectomy has been considered the procedure of choice [1]. However, in recent years, there has been growing interest in sublobar resections as potential alternatives to lobectomy, particularly in patients with small, peripheral, node-negative tumors.

The anatomy of the left upper lobe—divided into the upper division (segments I–III) and the lingula (segments IV and V)—favors segmental resections. Several authors have proposed that trisegmentectomy could be a valid alternative to lobectomy for tumors located in the upper division of the left lung without lingular involvement [2,3,4,5,6,7,8,9,10,11]. Previous studies have identified left upper lobe trisegmentectomy as an oncologically equivalent option to left upper lobectomy for tumors beyond the criteria established for segmentectomies [2,3,4,5,6,7,8,9,10,11,12].

Although some studies suggest that trisegmentectomy may offer comparable long-term survival to lobectomy, no randomized clinical trials have yet been conducted beyond the standard indications for segmentectomy. In recent years, several comparative studies evaluating these techniques have been published. Notably, a meta-analysis published in 2023 included five studies conducted between 2007 and 2022 [7]. Since then, four additional comparative studies have been published, reflecting the increasing interest in this topic [8,9,10,11].

Most previous studies have focused on analyzing overall survival, rather than examining recurrence patterns. However, several reports have suggested that lobectomy may be associated with a higher incidence of distant metastases compared to trisegmentectomy [9,10]. In the absence of randomized clinical trials, the findings from these retrospective studies should be interpreted with caution. While methodological biases may be present, the biological plausibility of these results should be carefully evaluated, as such findings may play a crucial role in guiding the surgical approach.

The aim of this meta-analysis is to evaluate whether trisegmentectomy offers non-inferior or even superior oncologic outcomes compared to lobectomy, with a particular focus on recurrence patterns.

## 2. Materials and Methods

This review adheres to the PRISMA (Preferred Reporting Items for Systematic Reviews and Meta-Analyses) 2020 guidelines. The protocol was registered in the PROSPERO international register of systematic reviews (registration number: CRD420251066445).

### 2.1. Inclusion Criteria

Studies included in this review were comparative analyses of left upper lobectomy versus left upper lobe trisegmentectomy. The outcomes of interest were recurrence (both locoregional and distant), morbidity, the length of hospital stay, and overall survival.

### 2.2. Identifying Relevant Studies

With the assistance of a documentation specialist, an electronic search strategy was developed in collaboration with members of the research team experienced in conducting systematic reviews. Online databases searched included MEDLINE and EMBASE via the OVIDSP interface, with a final search cutoff date of November 2024 (see Figure 1 for details of the search strategy).

### 2.3. Study Selection

Two reviewers independently assessed the relevance of the identified studies by screening titles and abstracts to determine eligibility for inclusion. Discrepancies were resolved through discussion and consensus. Full-text articles passing the initial screening were retrieved for further evaluation, including quality assessment, data extraction, and final inclusion decisions.

### 2.4. Quality Assessment

Two reviewers independently assessed the quality of the included studies in a non-blinded manner. Disagreements were resolved by consensus. The evaluation followed the criteria for cohort studies outlined in the Spanish version of the Critical Appraisal Skills Programme (CASPe: http://www.redcaspe.org (accessed on 1 December 2024)), excluding the first item (“Clearly focused research question”), which was a precondition for inclusion. The following criteria were assessed:Was the cohort recruited in an acceptable manner?Was the exposure accurately measured to minimize bias?Was the outcome accurately measured to minimize bias?Have the authors identified all important confounding factors?How precise are the results?Was the follow-up of subjects sufficiently long and complete?

### 2.5. Data Retrieval

Two reviewers independently extracted data using a predefined data extraction sheet. Extracted data included the following:Descriptive data: country, year of publication, sample size, mean age, sex distribution, and histological subtypes.Methodological data: strategies to reduce bias (e.g., propensity score matching), and the number of patients lost to follow-up.Outcome data: morbidity, local recurrence, distant recurrence, and overall recurrence. Recurrence was classified as local or distant; when studies reported mixed recurrences, these cases were included in both categories.

### 2.6. Measuring the Outcomes

Outcomes were analyzed according to predefined criteria. A quantitative approach was used to assess recurrence (locoregional and distant) based on the extracted data.

### 2.7. Statistical Analysis

To evaluate the comparative effectiveness of lobectomy and trisegmentectomy regarding recurrence, we performed meta-analyses using the metabin function from the R meta package (version 7.0). Odds ratios (ORs) were calculated as effect measures, using the Mantel–Haenszel method for fixed-effect analysis by default. Three separate meta-analyses were performed: (i) local recurrence, (ii) distant recurrence, and (iii) overall recurrence.

Additionally, a meta-analysis was conducted to evaluate differences in the length of hospital stay. In cases where only the median and interquartile range (IQR) were reported, and not the mean and standard deviation (SD), we assumed a normal distribution of hospitalization days. Under this assumption, the mean was approximated by the median, and the SD was estimated as IQR/1.35, based on the expected IQR for a standard normal distribution. For this analysis, the metacont function from the R meta package was used with default parameters [13].

In addition, for each analysis, we conducted two sensitivity studies. The first consisted of analyzing only the studies in which matching techniques were performed. Finally, a more comprehensive sensitivity analysis was performed using the leave-one-out approach to assess the robustness of the meta-analysis results. This method systematically excludes each individual study from the analysis and recalculates the pooled effect estimate to determine whether any single study disproportionately influences the overall results. The analysis was conducted using the metainf () function in the R (version 4.4.3) statistical software meta package. The Mantel–Haenszel method was used for binary outcomes (odds ratios) and the inverse variance method for continuous outcomes (mean differences). Heterogeneity was assessed using the restricted maximum likelihood estimator for tau^2^ and I^2^ statistics.

## 3. Results

The initial search identified 14 articles. Following the study selection process, nine studies were deemed original and relevant to the research question, which focused on comparing left upper lobectomy with left upper lobe trisegmentectomy. The study selection process is summarized in the PRISMA flowchart presented in Figure 2.

All included studies were retrospective comparative analyses, most of which were based on prospectively collected data. A summary of the nine included studies is provided in Table 1.

### 3.1. Methodological Quality

Three of the included studies did not use statistical methods to create comparable groups; therefore, the cohorts compared may not have been equivalent at baseline.

Among the remaining six studies, two included a heterogeneous non-lobectomy group that combined trisegmentectomies with other types of resections. These studies did not distinguish between trisegmentectomy and other procedures when reporting outcomes such as survival, recurrence, or morbidity, making it impossible to accurately assess the specific results of trisegmentectomy.

Table 2 presents the included studies and their methodological quality, including the variables used for propensity score matching.

### 3.2. Measured Outcomes

#### 3.2.1. Locoregional Recurrence

Locoregional recurrence (Figure 3) showed no significant differences between the lobectomy and trisegmentectomy groups. The study by Aprile reported a trend toward lower recurrence in the lobectomy group (6.7% vs. 12.9%), whereas the study by McAllister showed an opposite trend, favoring trisegmentectomy (10.8% vs. 3.6%). The meta-analysis confirmed no statistically significant difference in locoregional recurrence between the two surgical approaches. When the analysis was restricted to studies employing matching techniques, a clearer trend toward lower locoregional recurrence in the trisegmentectomy group was observed (OR 0.80; 95% CI: 0.56–1.14), although this did not reach statistical significance.

#### 3.2.2. Distant Recurrence

The meta-analysis demonstrated a significantly lower rate of distant recurrence in the trisegmentectomy group compared to the lobectomy group (Figure 4; OR 0.58; 95% CI: 0.41–0.82). This trend was consistent across most individual studies and reached statistical significance in the pooled analysis. Similar results were observed when the analysis was restricted to studies employing matching techniques.

#### 3.2.3. Overall Recurrence

The meta-analysis of overall recurrence (Figure 5) revealed a significantly lower risk of recurrence in the trisegmentectomy group (OR 0.65; 95% CI: 0.50–0.85). This difference remained significant when considering only matched studies (OR 0.63; 95% CI: 0.47–0.85).

#### 3.2.4. Morbidity and Length of Stay

The meta-analysis of morbidity (Figure 6) showed no significant difference between groups when all studies were included (OR 0.95). However, when analyzing matched studies, morbidity was significantly lower in the trisegmentectomy group (OR 0.73; 95% CI: 0.56–0.96). Regarding the length of hospital stay (Figure 7), both the overall meta-analysis and the subgroup of matched studies demonstrated a significantly shorter stay in the trisegmentectomy group.

### 3.3. Sensitivity Analysis

Appendix A present the results of the sensitivity analysis performed using the leave-one-out technique.

Regarding the recurrence analysis, for local recurrence, when individual studies were sequentially omitted, the odds ratio remained consistently non-significant, ranging from 0.7979 to 0.9477 (*p*-values: 0.2143–0.7721). For distant recurrence, the odds ratio remained stable between 0.5299 and 0.6393, with all *p*-values ≤ 0.0607. For global recurrence, the odds ratio ranged narrowly from 0.6190 to 0.6781 (*p*-values: 0.0016–0.0137), demonstrating minimal variability regardless of which study was excluded. The strongest effect was observed when the study by Tane (2024) [9] was omitted (OR: 0.6190, *p* = 0.0137), while the weakest effect remained highly significant when the study by Aguinagalde (2024) [10] was excluded (OR: 0.6764, *p* = 0.0074). Overall, no single study dominated the results, confirming the robustness of our findings for recurrence differences between surgical approaches.

Regarding morbidity, the odds ratio varied considerably from 0.7843 to 1.0546 (*p*-values: 0.0564–0.9942), with heterogeneity (I^2^) ranging from 0% to 66.8%. It is worth noting that excluding Soukiasian (2012) [3] resulted in a borderline significant reduction in morbidity (OR: 0.7843, *p* = 0.0564) with complete elimination of heterogeneity (I^2^ = 0%), suggesting this study may be a source of heterogeneity. Conversely, the inclusion of the study by Zhou (2022) [6] shifted the result toward no effect (OR: 1.0009, *p* = 0.9942), indicating that morbidity outcomes are more sensitive to individual study characteristics and should be interpreted with caution.

Finally, for the length of stay, the mean difference remained highly significant in all scenarios, ranging from −0.8890 to −1.2719 days (all *p*-values < 0.0001). The strongest effect was observed when the study by Zhou (2022) [6] was omitted (MD: −1.2719 days), with complete elimination of heterogeneity (I^2^ = 0%), while the weakest but still highly significant effect occurred when the study by Soukiasian (2012) [3] was excluded (MD: −0.8890 days). These results demonstrated robust and consistent findings across all leave-one-out scenarios.

## 4. Discussion

Our meta-analysis confirms, in line with previous studies, that trisegmentectomy does not confer a worse prognosis than left upper lobectomy for early-stage non-small-cell lung cancer (NSCLC). However, unlike earlier works, we specifically focused on recurrence patterns and observed that patients undergoing left upper lobectomy have a higher likelihood of distant recurrence compared to those treated with trisegmentectomy.

While Bayfield et al. [7] conducted a systematic review with meta-analysis comparing lobectomy and trisegmentectomy in 2023, only five studies were included, of which only two employed propensity score matching to balance the groups. Moreover, their analysis centered on survival outcomes rather than recurrence patterns.

The observed increased tendency for distant metastases in the lobectomy group may partly reflect inherent biases in the included studies. Six of the nine selected studies used statistical techniques to create comparable groups, yet the variables used for matching varied considerably. Age, sex, tumor size, histology, and lung function were commonly matched, but other important factors such as SUVmax—a recognized predictor of recurrence and prognosis after lung cancer resection [14,15]—were included only in the studies by Nishikubo and Aguinagalde [8,10]. Tumor density was matched solely in these two and Zhou’s study, while central versus peripheral tumor location—a potentially influential variable—was not controlled for in any study. Although statistical heterogeneity was low (I^2^ = 0%), potential sources of clinical heterogeneity should be acknowledged. Studies varied in surgical approaches (thoracoscopic vs. open techniques), patient populations (mean age ranged from X to Y years), and discharge criteria protocols.

Additionally, although propensity score matching mitigates confounding, the non-randomized nature of the studies implies that unmeasured confounders (e.g., tumor location, surgeon experience) may still influence the results [16].

Beyond methodological considerations, the biological plausibility of the differing recurrence patterns warrants attention. The surgical technique may contribute to this difference: most surgeons performing fissureless trisegmentectomy ligate the vein before the arteries [17], whereas left upper lobectomy typically involves ligation of the artery prior to the veins. Several studies have documented increased dissemination of tumor cells into the bloodstream when the artery is ligated before the veins, correlating with poorer overall and cancer-specific survival, especially in early-stage disease [18,19,20,21]. This mechanism may underlie the higher frequency of distant metastases observed in the lobectomy group.

Moreover, segmentectomy appears to be less biologically aggressive and better preserves patients’ immunological and nutritional status compared to lobectomy. Preoperative immunological and nutritional status has been linked to prognosis in NSCLC patients [22], and postoperative differences in these parameters may explain variations in mortality risk between segmentectomy and lobectomy groups. A recent study identified perioperative changes in the prognostic nutritional index as a survival determinant in surgically treated lung cancer patients [23]. Our analysis suggests that morbidity and hospital stay, factors that could also impact recurrence risk, are lower in the trisegmentectomy group.

Despite the absence of randomized controlled trials in our meta-analysis, the randomized JCOG0802 trial reported similar findings, with distant metastases without locoregional recurrence being more frequent in the lobectomy group compared to the segmentectomy group (2.5% vs. 1.3%) [24]. Subsequent analyses of JCOG0802 suggest that preservation of immunonutritional status may explain why segmentectomy achieved non-inferior survival despite a higher rate of locoregional recurrence [23].

Taken together, current evidence indicates a higher rate of distant metastases following left upper lobectomy compared to trisegmentectomy. While this meta-analysis cannot definitively rule out bias inherent to non-randomized studies, the existence of biologically plausible mechanisms—supported by indirect evidence from JCOG0802—highlights the need for further research. We advocate for the design of a randomized clinical trial comparing lobectomy and trisegmentectomy in tumors larger than 2 cm, with particular attention to recurrence patterns. Additionally, future studies comparing lobectomy and sublobar resections should incorporate a detailed analysis of recurrence patterns.

### Limitations

The included studies were non-randomized, and despite the use of propensity score matching, not all relevant confounders were accounted for. Important variables such as tumor location and surgeon experience were not measured and may have influenced the outcomes.

Although outcome variables were relatively homogeneous across studies, clinical stage distributions varied. For example, Aguinagalde et al. included patients beyond stage I, whereas other studies focused primarily on early stages. The lack of access to individual patient data precluded subgroup analyses by stage, limiting the precision of conclusions. Variability also existed in the variables selected for matching across studies.

Furthermore, the immunological and nutritional status of patients was not assessed, nor were surgical techniques comprehensively described, limiting mechanistic interpretations. Therefore, the hypothesis that lobectomy may lead to greater distant dissemination requires confirmation through higher-quality prospective trials.

## 5. Conclusions

In conclusion, our findings suggest that trisegmentectomy and left upper lobectomy are associated with different recurrence patterns, with a higher tendency for distant dissemination observed after lobectomy. This finding warrants validation in prospective clinical trials.

## Figures and Tables

**Figure 1 jcm-14-04385-f001:**
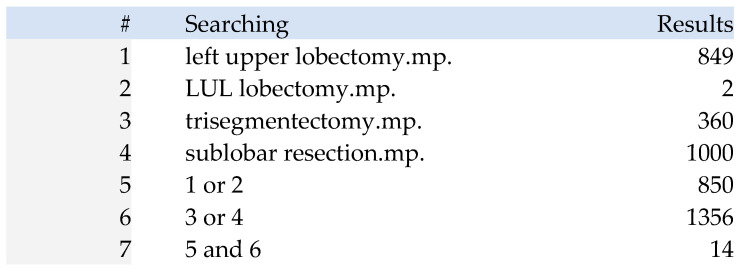
Search strategy formulation (with keywords and Boolean operators).

**Figure 2 jcm-14-04385-f002:**
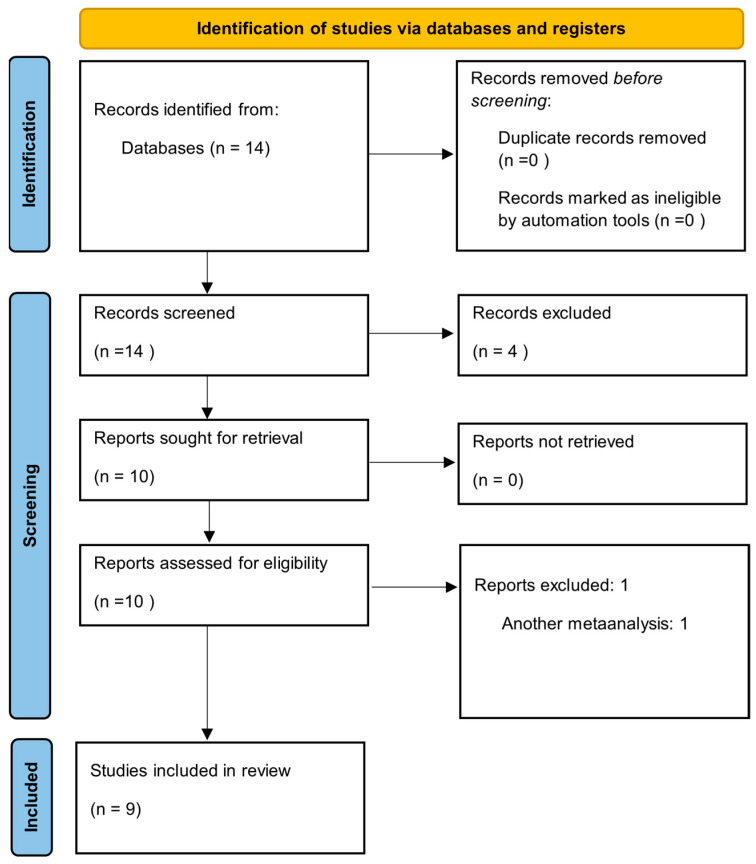
The Preferred Reporting Items for Systematic Reviews and Meta-Analyses (PRISMA) flowchart.

**Figure 3 jcm-14-04385-f003:**
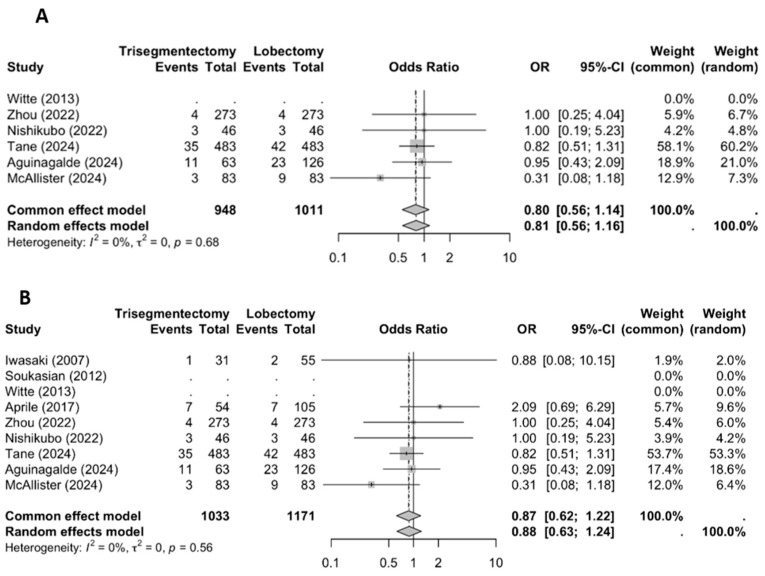
A locorregional recurrence forest plot ((**A**) all studies and (**B**) matched studies) [2,3,4,5,6,8,9,10,11].

**Figure 4 jcm-14-04385-f004:**
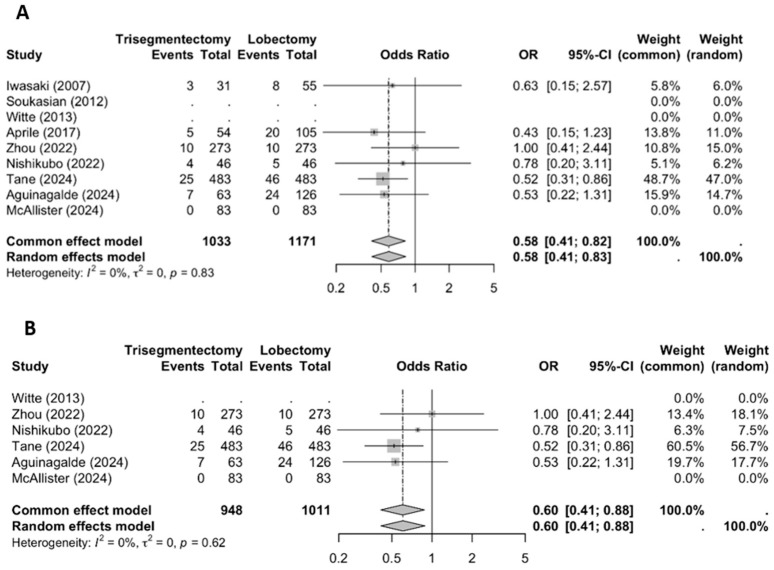
A distant recurrence forest plot ((**A**) all studies and (**B**) matched studies) [2,3,4,5,6,8,9,10,11].

**Figure 5 jcm-14-04385-f005:**
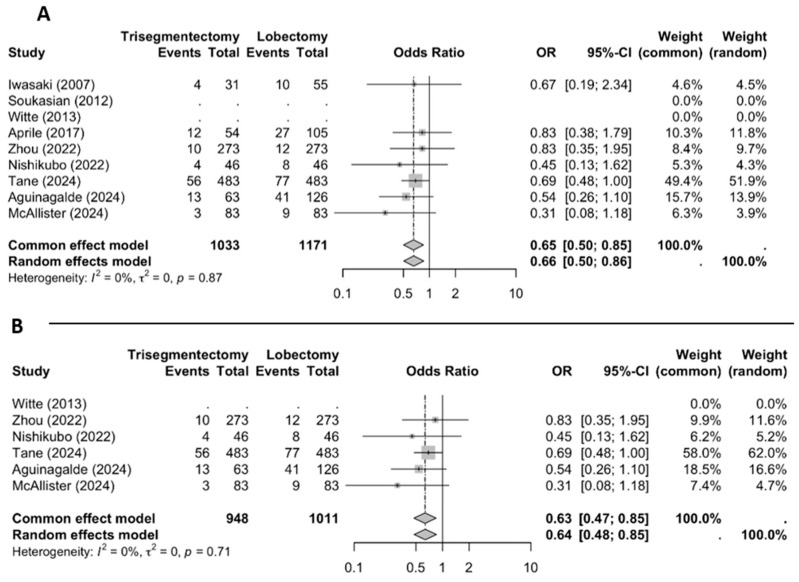
An overall recurrence forest plot ((**A**) all studies and (**B**) matched studies) [2,3,4,5,6,8,9,10,11].

**Figure 6 jcm-14-04385-f006:**
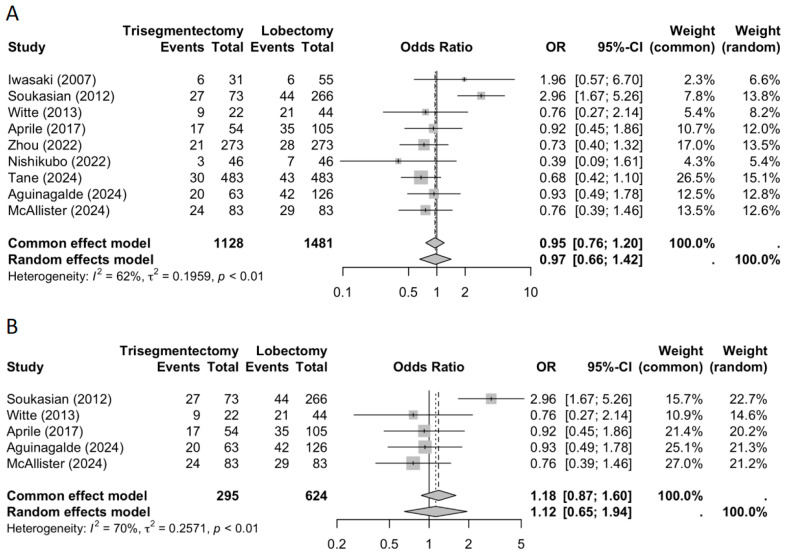
A morbidity forest plot ((**A**) all studies and (**B**) matched studies) [2,3,4,5,6,8,9,10,11].

**Figure 7 jcm-14-04385-f007:**
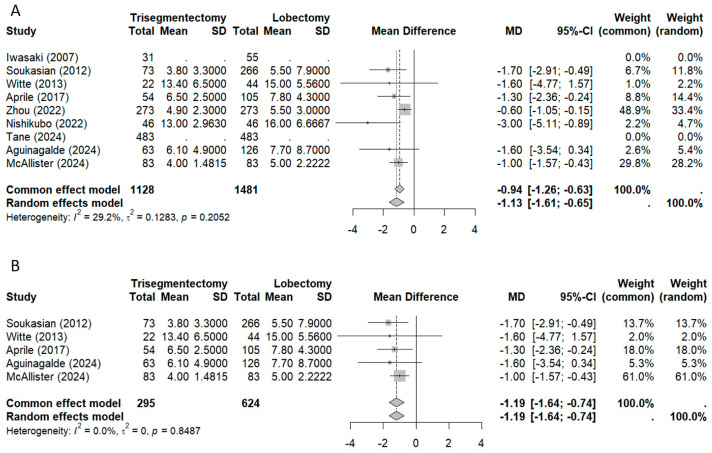
The length of stay forest plot ((**A**) all studies and (**B**) matched studies) [2,3,4,5,6,8,9,10,11].

**Table 1 jcm-14-04385-t001:** Summary of included studies.

Author	Year	Country	Design	Surgery	Stage	Adenoca	Female
Iwasaki	2007	Japan	RAoPD	LUL vs. TRI	Stage IA	%75.6	%43
Soukasian	2012	EEUU	RAoPD	LUL vs. TRI	Stage I	-----	------
Witte	2013	Germany	Retrospective	LUL vs. TRI or lingula	pStage I-IIIA	%34.8	%27.3
Aprile	2017	Italy	RAoPD	LUL vs. TRI or lingula	Stage IA-IIA	%61.6	%20.8
Zhou	2022	China	RAoHR	LUL vs. TRI	cStage I	%96.3	%61.3
Nishikubo	2022	Japan	RAoHR	LUL vs. TRI	Stage I-IIB	%71.7	%31.5
Tane	2024	Japan	RAoPD	LUL vs. TRI	cStage I	%87.8	%49.5
Aguinagalde	2024	Spain	RAoPD	LUL vs. TRI	All	-------	--------
McAllister	2024	EEUU	RAoPD	LUL vs. TRI	Stage cIA-IIA	%78.2	%65.5

RAoPD: retrospective analysis of prospective data. RAoHR: retrospective analysis of hospital records. LUL: left upper lobectomy. TRI: trisegmentectomy.

**Table 2 jcm-14-04385-t002:** Methodological quality of studies.

Author	Patients	PM	Variables for PM	Quality	Issues
Iwasaki	86	No	-	Low	No comparable groups
Soukasian	339	No	-	Low	No comparable groups
Witte	66	Yes	Histology, pN, tumor diameter, age	Medium	TRI and lingulectomy are mixed
Aprile	159	No	-	Low	No comparable groups
Zhou	546	Yes	Age, sex, CT appearance, pathologic type, tumor size, pathologic stage	High	
Nishikubo	92	Yes	Age, sex, smoking history, histology, SUVmax, cT, PFT, GGO.	High	
Tane	966	Yes	Age, sex, PS, CEA, PFT, tumor size	Medium	The number of patients for each type of left upper lobe segmentectomy is not specified
Aguinagalde	189	Yes	Tumor size, tumor density, surgical access, cN, pN, pTNM, histological subtype, PFT, stroke, diabetes	High	
McAllister	166	Yes	Age, sex, CCI, PFT, smoking history, cT	High	

SUVmax: maximum standardized uptake value. PFT: pulmonary function tests. GGO: ground-glass opacities. pN: pathological N factor. cT: clinical T factor. PS: performance status. CEA: carcinoembryonic antigen.

## Data Availability

The original contributions presented in this study are included in the article/Appendix A. Further inquiries can be directed to the corresponding author.

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
