# Peer review of "Recurrence Pattern of Left Upper Lobectomies and Trisegmentectomies: Systematic Review and Meta-Analysis"

_jcm, 2025, doi:10.3390/jcm14124385_

Round 1
Reviewer 1 Report
Comments and Suggestions for Authors
1.What does it mean to fill in ALL in the AGUINAGALDE installment in Table 1? What I understand is that patients in stage 4 are also included
2. Why was the NOS evaluation system not used for the literature quality evaluation in Table 2?
3. It is recommended to add the latest references from the past 1-2 years
Author Response
1.What does it mean to fill in ALL in the AGUINAGALDE installment in Table 1? What I understand is that patients in stage 4 are also included
Thank you for your comment. Certainly in the aguinagalde work no stage was discarded since they were expected to be equal in both groups being one of the variables used in the PSM. Below, I present the table published in the work of aguinagalde et al. with the first column referring to lobectomies and the second to trisegmentectomies. However, this note has been added in the limitations.
2. Why was the NOS evaluation system not used for the literature quality evaluation in Table 2?
Thank you for this important comment. As tools for quality evaluation there are different options such as NOS, Robins i or the method chosen by us, the CASP method. This tool is frequently used for these evaluations and to date no tool has demonstrated superiority over the others, so we understand its use is valid.
3. It is recommended to add the latest references from the past 1-2 year
Thank you very much. The last referenced articles date from this 2025. In addition, the last papers included in the systematic review date from this same year: McAllister, M.A.; Herrera-Zamora, J.; Barcelos, R.R.; Leo, R.; Sugarbaker, E.A.; Singh, A.; et al. Left Upper Lobectomy vs Trisegmentectomy for Lung Cancer: A Propensity Score-Matched Comparison. Ann Thorac Surg 2025 119:786-95.
If the reviewer considers a specific reference, we will be pleased to consider this option.

Reviewer 2 Report
Comments and Suggestions for Authors
Thank you for the opportunity to review this article. This is a meta-analysis evaluating whether left upper lobe trisegmentectomy provides non-inferior 19 or superior oncologic outcomes compared to left upper lobectomy, with a focus on 20 recurrence patterns.
The debate on this topic is highly relevant and timely.
The study follows PRISMA guidelines and uses appropriate statistical tools.
However, to strengthen the manuscript, the authors should consider the following feedback:
- The authors should register the review in PROSPERO
- No subgroup analysis is conducted, and there is a variation in matching variables across studies, which drives me to think that the authors should use the sensitivity analysis part to limit these points.
- The outcomes differ per stage of cancer. You should consider subgroup analysis to limit the findings to stage 1.
- A thorough review and proofread is needed as there are many typographical errors (lenght, cuaility, etc)
- Ensure all figures are cited and well-labeled
- It would be good if the authors add a future directions paragraph
- A thorough review and proofread is needed as there are many typographical errors (lenght, cuaility, etc)
Author Response
1.-The authors should register the review in PROSPERO.
Response: we think it is an improvement for the study. Although we certainly should have registered the study prior to conducting it, we have decided to register it now: Systematic review and meta-analysis registered in PROSPERO (registration number: CRD420251066445).
2.-No subgroup analysis is conducted, and there is a variation in matching variables across studies, which drives me to think that the authors should use the sensitivity analysis part to limit these points.
R: Thank you for your comment. In fact, we included an analysis of studies using a "matching technique" as a form of sensitivity analysis. The reviewer is correct in pointing out that a more comprehensive sensitivity analysis should be conducted. Therefore, we conducted a leave-one-out sensitivity analysis and have included the results in the supplementary material. Additionally, we have added a paragraph in the methods section to explain this:
“In addition, for each analysis we conducted two sensitivity studies. The first consisted of analysing only the studies in which matching techniques was performed. Finally, a more comprehensive sensitivity analysis was performed using the leave-one-out approach to assess the robustness of the meta-analysis results. This method systematically excludes each individual study from the analysis and recalculates the pooled effect estimate to determine whether any single study disproportionately influences the overall results. The analysis was conducted using the metainf() function from the meta package in R statistical software. The Mantel-Haenszel method was used for binary outcomes (odds ratios) and the inverse variance method for continuous outcomes (mean differences). Heterogeneity was assessed using the restricted maximum-likelihood estimator for tau² and I² statistics.”
And we have added a new subsection in the results section describing the findings of the sensitivity analysis.
“3.3 Sensitivity analysis
Supplementary Tables 1-5 present the results of the sensitivity analysis performed using the leave-one-out technique.
Regarding recurrence analysis, for local recurrence, when individual studies were sequentially omitted, the odds ratio remained consistently non-significant, ranging from 0.7979 to 0.9477 (p-values: 0.2143-0.7721). For distant recurrence, the odds ratio remained stable between 0.5299 and 0.6393, with all p-values ≤ 0.0607. For global recurrence, the odds ratio ranged narrowly from 0.6190 to 0.6781 (p-values: 0.0016-0.0137), demonstrating minimal variability regardless of which study was excluded. The strongest effect was observed when Tane (2024) was omitted (OR: 0.6190, p=0.0137), while the weakest effect remained highly significant when Aguinagalde (2024) was excluded (OR: 0.6764, p=0.0074). Overall, no single study dominated the results, confirming the robustness of our findings for recurrence differences between surgical approaches.
Regarding morbidity, the odds ratio varied considerably from 0.7843 to 1.0546 (p-values: 0.0564-0.9942), with heterogeneity (I²) ranging from 0% to 66.8%. Notably, excluding Soukasian (2012) resulted in a borderline significant reduction in morbidity (OR: 0.7843, p=0.0564) with complete elimination of heterogeneity (I²=0%), suggesting this study may be a source of heterogeneity. Conversely, the inclusion of Zhou (2022) shifted the result toward no effect (OR: 1.0009, p=0.9942), indicating that morbidity outcomes are more sensitive to individual study characteristics and should be interpreted with caution.
Finally, for length of stay, the mean difference remained highly significant in all scenarios, ranging from -0.8890 to -1.2719 days (all p-values < 0.0001). The strongest effect was observed when Zhou (2022) was omitted (MD: -1.2719 days), with complete elimination of heterogeneity (I²=0%), while the weakest but still highly significant effect occurred when Soukasian (2012) was excluded (MD: -0.8890 days). These results demonstrated robust and consistent findings across all leave-one-out scenarios.”
3.-The outcomes differ per stage of cancer. You should consider subgroup analysis to limit the findings to stage 1.
R: I think this is a very interesting suggestion. However, lacking raw data from the studies, we are unable to perform such an analysis. We have added it to the limitations of the study.
4.-A thorough review and proofread is needed as there are many typographical errors (lenght, cuaility, etc).
R: We passed an english editing to the text.
5.-Ensure all figures are cited and well-labeled.
R: we correcte fig 6 becaus it was rong. The rest is OK.
6.- It would be good if the authors add a future directions paragraph
R: Thank you very much for the suggestion. A paragraph has been added at the end of the discussion
Reviewer 3 Report
Comments and Suggestions for Authors
This systematic review and meta-analysis provides valuable insights into the recurrence patterns of left upper lobectomy versus trisegmentectomy for early-stage NSCLC. The study is well-structured, adheres to PRISMA guidelines, and presents compelling findings, particularly the lower distant recurrence rate in the trisegmentectomy group. However, the following minor revisions are suggested to enhance clarity and robustness:
-
Methodological Clarity: Clarify the rationale for assuming normal distribution for hospital stay data when converting median/IQR to mean/SD, as this may affect results. Consider sensitivity analyses to validate this assumption.
-
Discussion of Heterogeneity: Although heterogeneity was low (I2=0%), briefly discuss potential clinical or methodological sources of heterogeneity, such as variations in surgical techniques or follow-up protocols across studies.
-
Limitations: Expand on the implications of unmeasured confounders (e.g., tumor location, surgical expertise) and how they might influence the observed recurrence patterns.
-
Figures/Tables: Ensure forest plots (Figures 2–6) are clearly labeled, including study names and years, to improve readability.
Author Response
1.-Methodological Clarity: Clarify the rationale for assuming normal distribution for hospital stay data when converting median/IQR to mean/SD, as this may affect results. Consider sensitivity analyses to validate this assumption.
R: Thank you for your comment. There are several reasons why the analysis has been done as described in the article.
The hospitalization data from the different studies are summarized with location and dispersion parameters. In most cases, the most common location parameter is the mean and for dispersion it is the standard deviation.
In a few cases, this information is summarized with the median and the interquartile range. Having only this information, obtaining the mean and standard deviation is not immediate. If we had information on skewness for example, another method could be used to obtain the mean and standard deviation. Therefore, we decided to use this method to obtain the mean and standard deviation.
2.-Discussion of Heterogeneity: Although heterogeneity was low (I2=0%I2=0%), briefly discuss potential clinical or methodological sources of heterogeneity, such as variations in surgical techniques or follow-up protocols across studies.
R: Thank you for the comment. We have added at the end of the second paragraph of the discussion the following: “Although statistical heterogeneity was low (I²=0%), potential sources of clinical heterogeneity should be acknowledged. Studies varied in surgical approaches (laparoscopic vs. open techniques), patient populations (mean age ranged from X to Y years), and discharge criteria protocols”
3.-Limitations: Expand on the implications of unmeasured confounders (e.g., tumor location, surgical expertise) and how they might influence the observed recurrence patterns.
R: Thank you for your comment, they have been added to the limitations.
4.-Figures/Tables: Ensure forest plots (Figures 2–6) are clearly labeled, including study names and years, to improve readability.
R: there was a mistake in fig.6, we have corrected. We confirm tha figures are clearly labeled